# VIPO-R1: CULTIVATING VIDEO REASONING IN MLLMs VIA VERIFIER-GUIDED ITERATIVE POLICY OPTIMIZATION

## ABSTRACT

Applying Reinforcement Learning (RL) to Multimodal Large Language Models (MLLMs) shows significant promise for complex video reasoning. However, popular Reinforcement Fine-Tuning (RFT) methods, such as outcome-based Group Relative Policy Optimization (GRPO), are limited by data preparation bottlenecks (e.g., noise or high cost) and exhibit unstable improvements in the quality of long chain-of-thoughts (CoTs) and downstream performance. To address these limitations, we propose **VIPO-R1**, a **V**erifier-guided **I**terative **P**olicy **O**ptimization method designed to gradually enhance MLLMs' ability to generate long-term reasoning chains for challenging VideoQA. The core component is the Rollout-Aware Verifier, positioned between the GRPO and Direct Preference Optimization (DPO) training phases to form the GRPO-Verifier-DPO training loop. This verifier leverages small LLMs as a judge to assess the reasoning logic of rollouts, enabling the construction of high-quality contrastive data, including reflective and contextually consistent CoTs. These curated preference samples drive the efficient DPO stage (7x faster than GRPO), leading to marked improvements in reasoning chain quality, especially in terms of length and contextual consistency. This training loop benefits from GRPO's expansive search and DPO's targeted optimization. Experimental results demonstrate: 1) Faster and more effective optimization compared to standard GRPO variants, yielding superior performance; 2) Our trained models exceed the direct inference of large-scale instruction-tuned Video-LLMs, producing long and contextually consistent CoTs on diverse video reasoning tasks; and 3) Our model with one iteration outperforms powerful MLLMs (e.g., Kimi-VL) and thinking models (e.g., Video-R1), highlighting its effectiveness and stability.

## 1 INTRODUCTION

Complex reasoning problems across various domains are often effectively tackled by large models via generating long Chain-of-Thoughts (CoTs) (Wei et al., 2023; Zhang et al., 2024d; Zelikman et al., 2022; Li et al., 2025e), which has demonstrated considerable success in multimodal settings, particularly for challenging tasks like visual math and complex image-text reasoning (Wang et al., 2025d; Dong et al., 2025; Team et al., 2025; Wu et al., 2025; Xu et al., 2025; Xiang et al., 2024). The capacity of Large Multimodal Models (LMMs) for long-form CoT reasoning is largely driven by Reinforcement Fine-Tuning (RFT), which integrates Supervised Fine-Tuning (SFT) with long-form CoT data and employs online reinforcement learning algorithms (Tan et al., 2025; Schulman et al., 2017; Rafailov et al., 2024; Yu et al., 2025; Gupta et al., 2025; Wu et al., 2024b; Tang et al., 2025; Team, 2025), such as the Group Relative Policy Optimization (GRPO) (Shao et al., 2024) method. Inspired by the success of DeepSeek-R1 (DeepSeek-AI et al., 2025), Skywork R1V (Chris et al., 2025), and Vision-R1 (Huang et al., 2025), researchers (Li et al., 2025b; Feng et al., 2025; Zhang et al., 2025c) are actively exploring effective strategies to enable Multimodal Large Language Models (MLLMs) to generate coherent and extensive reasoning chains for challenging VideoQA tasks.

However, activating the long-form reasoning of MLLMs for video understanding faces challenges:

- *Data Preparation Bottleneck*: Employing Long-CoTs video datasets for cold starting (e.g., Video-R1) is hindered by the high cost of manual annotation and noise from automatic methods.

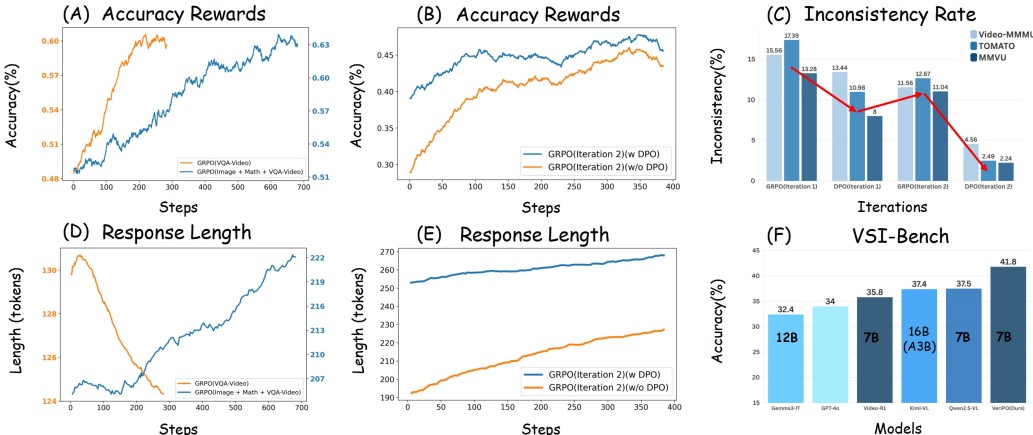

Figure 1: Figures (A, D): Initial GRPO training with different data types shows only utilizing Video-QA data decreases response length. Figures (B, E): Continual GRPO training with/without Verifier-guided DPO (VIPO-R1) demonstrates VIPO-R1 improves accuracy and response length. Figure (C): Inconsistency rate (thinking vs. final answer) at different stages reveals our method lowers contextual inconsistency of long CoTs while GRPO increases it. Figure (F): Performance on challenging video reasoning dataset VSI-Bench (Yang et al., 2024) shows VIPO-R1 (trained with Qwen2.5-VL-7B) outperforms strong LMMs including GPT-4o (Hurst et al., 2024), Video-R1 (Feng et al., 2025), and Kimi-VL (Team et al., 2025).

- *Unstable performance improvement*: GRPO training on video datasets can easily lead to decreases in reasoning length (Figure (D)) and model performance (Video-R1 vs. Qwen2.5-VL in Figure (F)). The model after GRPO training often performs worse than the original direct-answer model.

- *Inconsistency between reasoning and answers*: GRPO training often results in a misalignment between the reasoning chain and the final answer, leading to logically inconsistent outcomes such as "correct answers derived from flawed reasoning", as illustrated in Figure (C). This issue, which undermines interpretability and limits performance, stems from GRPO's reliance on final answer-based rewards without intermediate supervision.

To address these limitations, we propose **VIPO-R1**, an online rollout-aware **V**erifier-guided **I**terative **P**olicy **O**ptimization algorithm designed to progressively enhance the long-form reasoning capability of MLLMs on video understanding tasks. Unlike methods that rely on large-scale Long-CoTs video datasets for cold-start training, VIPO-R1 employs reinforcement learning to incrementally cultivate extended reasoning skills in MLLMs. A central component of our framework is the rollout-aware Verifier, which bridges the GRPO and DPO training phases to form a closed-loop GRPO–Verifier–DPO cycle. This verifier utilizes small LLMs to evaluate the reasoning quality and contextual coherence of generated CoTs from the online RL stage. Based on this assessment, it intelligently selects high-quality contrastive samples from online rollouts to construct logically consistent and reflective reasoning chains. These samples are then used in an efficient DPO stage, which we empirically found to be 7× faster than GRPO (see Section C.1) and more effective at refining reasoning paths. Additionally, the verifier progressively prunes simple examples that the model has already mastered, which accelerates training and ensures a focus on more challenging instances. This filtering mechanism allows the policy optimization to effectively combine the targeted refinement of DPO with the broad exploration capability of GRPO. To further diversify the logical reasoning paths learned by the model, our training incorporates a mixture of diverse VideoQA datasets, supplemented with high-quality image and textual math datasets during the initial phase.

We conduct extensive experiments on five video reasoning and long video understanding benchmarks, e.g., VSI-Bench (Yang et al., 2024) and Video-MME (Fu et al., 2024). Our experimental results show that **VIPO-R1** achieves consistent and significant performance improvements and outperforms larger MLLMs and powerful RFT models Video-R1 and Kimi-VL-Thinking (Team et al., 2025). It highlights the effectiveness and stability of VIPO-R1 in cultivating the long-form video reasoning ability of MLLMs. Compared to RFT with the long-CoTs dataset as a cold start, our approach consistently generates longer responses and improves the quality of generated long CoTs, e.g., contextual consistency and low repetition. Our contributions can be summarised as follows:

- We propose **VIPO-R1**, a novel Verifier-guided Iterative Policy Optimization algorithm designed to improve the long-form reasoning capability of MLLMs. The method enhances rollout data utilization via the embedded Verifier system and efficient DPO, enabling the model to realize improvement via effective learning from its online running experience.

- The rollout-aware Verifier analyzes and refines generated rollout data into high-quality, reflective contrastive samples, which are essential for continuously improving the model's long-form reasoning capability and logical consistency during the DPO training stage.

- Experimental results demonstrate that VIPO-R1 significantly improves long-form reasoning performance on challenging video QA tasks. Our trained models consistently generate long and accurate reasoning chains, outperforming direct-answer models (like Qwen2.5-VL-7B), GRPO baseline and RL-trained thinking models (including Video-R1, Kimi-VL-Thinking-16A3B) - gaining +4.0% on VSI-Bench over Qwen2.5-VL-7B and +3.6% on TOMATO over GRPO baseline.

## 2 RELATED WORK

**Large Multimodal Models for Video Reasoning**  Video reasoning is the core capability of Large Multimodal Models (LMMs), enabling understanding of interactions, dependencies, and inference over dynamic content (Li et al., 2024c; 2025d; Zhang et al., 2025c; Zheng et al., 2025). Specifically, spatial reasoning models object relationships and scene layouts within frames, while temporal reasoning captures motion, causality, and sequence across frames (Ouyang, 2025; Daxberger et al., 2025; Ray et al., 2025; Liu et al., 2025b). Early Video-LLMs focused on short videos using pre-trained image (Dosovitskiy et al., 2021; Oquab et al., 2024; Radford et al., 2021) or video encoders (Arnab et al., 2021; Liu et al., 2021; Neimark et al., 2021) with frozen language models (Dai et al., 2023; Li et al., 2022; 2024b; Maaz et al., 2024; Zhang et al., 2023). Recent efforts target long-form video understanding with complex temporal and multimodal reasoning (Fei et al., 2024; Feng et al., 2025; Zhang et al., 2025c; Zheng et al., 2025; Liu et al., 2025a; Chen et al., 2025c; Liu et al., 2025d). To handle long contexts, methods adopt hierarchical temporal attention and larger context windows (Liu et al., 2025a; Wei et al., 2025), or compress visual inputs via event-level abstraction (Zhang et al., 2025c; Chen et al., 2024). Recent multimodal fusion integrates audio and motion cues for improved understanding in videos (Chen et al., 2025c; Zhao et al., 2025a; Liu et al., 2025e). Reinforcement learning guides perception and reasoning, aiding in interpretability and intent modeling (Deng et al., 2025; Liu et al., 2025d;c). Recent work explores structured outputs, intention-driven attention, and stepwise reasoning (Chen et al., 2025c; Yang et al., 2025; Huang et al., 2025; Peng et al., 2025) for fine-grained grounding and spatiotemporal segmentation.

**Reinforcement Learning for Multimodal Reasoning**  Reinforcement learning (RL) has become a pivotal approach for aligning LLMs and LMMs with complex reasoning objectives. Foundational policy optimization algorithms, such as Proximal Policy Optimization (PPO), Direct Preference Optimization (DPO), and Group Relative Policy Optimization (GRPO), have been instrumental in this domain (Schulman et al., 2017; Rafailov et al., 2024; Shao et al., 2024). Further advancements have enhanced training stability and efficiency (Yu et al., 2025; Gupta et al., 2025; Wu et al., 2024b; Tang et al., 2025). A critical challenge in popular RL training is the "cold start" problem, where initializing models without prior guidance can lead to suboptimal performance. To mitigate this, Reinforcement Fine-Tuning (RFT) has been proposed, wherein models undergo preliminary SFT on curated datasets to stabilize subsequent RL training phases (Liu et al., 2025e; Zhang et al., 2024c; Tan et al., 2025; Shi et al., 2025; Chen et al., 2025a; Li et al., 2025b; Wang et al., 2025b; Luo et al., 2025; Wang et al., 2025d; Xing et al., 2025). Additionally, some verifiers, designed to assess and guide the quality of generated outputs, have proven beneficial. These verifiers assist in filtering and selecting high-quality training samples, thereby enhancing the efficiency and effectiveness of the training process (Chen et al., 2025c; Zhao et al., 2025a; Sun et al., 2025; Wang et al., 2024).

## 3 PRELIMINARY

**Direct Preference Optimization (DPO)**  DPO (Rafailov et al., 2024) optimizes a policy $\pi_\theta$ to prefer a response $y_+$ over $y_-$ for a given input $x$, with regularization from a reference model $\pi_{\text{ref}}$. The core loss function is:

$$\mathcal{L}_{\text{DPO}}(\pi_\theta; \pi_{\text{ref}}) = -\mathbb{E}_{(x, y_+, y_-) \sim \mathcal{D}} \left[ \log \sigma \left( \beta \log \frac{\pi_\theta(y_+ \mid x)}{\pi_{\text{ref}}(y_+ \mid x)} - \beta \log \frac{\pi_\theta(y_- \mid x)}{\pi_{\text{ref}}(y_- \mid x)} \right) \right], \quad (1)$$

where $\sigma(\cdot)$ is the sigmoid function, $\beta > 0$ is a temperature parameter, and $\mathcal{D} = \{(x, y_+, y_-)\}_{i=1}^N$ is a static dataset of comparisons sampled from human preference distribution. This can be interpreted as minimizing the binary cross-entropy between a pairwise preference label and the log odds induced by the policy relative to the reference. This approach is a *targeted and fast* optimization for models.

**Group Relative Policy Optimization (GRPO)**   For a given input $q$, the model generates a group of $G$ responses $\{y_1, y_2, \ldots, y_G\}$ sampled from the current policy $\pi_\theta$. Each response $y_i$ is assigned a reward $r(y_i)$, typically derived from human feedback or automated evaluation metrics. Following outcome supervision method, the group mean reward $\mu$ and standard deviation $\sigma$ are computed to obtain the advantage score:

$$\mu = \frac{1}{G} \sum_{i=1}^G r(y_i), \quad \sigma = \sqrt{\frac{1}{G} \sum_{i=1}^G (r(y_i) - \mu)^2}, \quad \hat{A}_{i,t} = \frac{r(y_i) - \mu}{\sigma}. \quad (2)$$

With the score computed, GRPO (Shao et al., 2024) updates the policy by maximizing the following objective:

$$\mathcal{L}_{\text{Advantage}}(\pi_\theta) = \mathbb{E}_{q \sim \mathcal{P}(Q), \{y_i\}_{i=1}^G \sim \pi_{\theta_{\text{ref}}}(y_{i,t} \mid q, y_{i,<t})}$$

$$\frac{1}{G} \sum_{i=1}^G \frac{1}{|y_i|} \sum_{t=1}^{|y_i|} \left\{ \min \left[ \frac{\pi_\theta(y_{i,t} \mid q, y_{i,<t})}{\pi_{\theta_{\text{ref}}}(y_{i,t} \mid q, y_{i,<t})} \hat{A}_{i,t}, \ \text{clip} \left( \frac{\pi_\theta(y_{i,t} \mid q, y_{i,<t})}{\pi_{\theta_{\text{ref}}}(y_{i,t} \mid q, y_{i,<t})}, 1 - \epsilon, 1 + \epsilon \right) \hat{A}_{i,t} \right] \right\}$$
$$(3)$$

$$\mathcal{L}_{\text{GRPO}}(\pi_\theta) = \mathcal{L}_{\text{Advantage}}(\pi_\theta) - \beta \, \text{D}_{\text{KL}} \left[ \pi_\theta \| \pi_{\text{ref}} \right] \quad (4)$$

where $\epsilon$ is a hyperparameter controlling the clipping range, $\beta$ is the temperature parameter, and $\pi_{\theta_{\text{ref}}}$ is the policy before the update. This approach allows the model to focus on generating responses that are relatively better within a group, promoting *wide yet slow* exploration in the generation space.

## 4 VIPO-R1: VERIFIER-GUIDED ITERATIVE POLICY OPTIMIZATION

### 4.1 OVERVIEW

We introduce VIPO-R1, an iterative policy optimization approach specifically designed to enhance the long reasoning capability of Video-LLMs. The method follows an iterative process: 1) *Initial Policy Exploration:* We first apply GRPO to the instruction-tuned Qwen2.5-VL, utilizing diverse accuracy rewards tailored for various video task output formats. 2) *Sample Curation with Verifier:* A Verifier component analyzes the GRPO rollouts to produce high-quality, long reasoning paths that lead to accurate answers as positive (chosen) samples. It also selects challenging, incorrect reasoning paths as hard negative (rejected) samples. 3) *Policy Refinement with DPO*: These curated contrastive samples are then used to fine-tune the model via DPO. The DPO efficiently refines the model's policy, encouraging the generation of better reasoning paths in a controllable direction.

### 4.2 GRPO

Following the GRPO algorithm from DeepSeek-R1 (DeepSeek-AI et al., 2025), we employ two types of rewards: accuracy and format. The accuracy reward $r_a$ is scaled within the range $[0, 1]$, while the format reward $r_f$ is bounded within $[0, 0.5]$. The calculation of accuracy reward $r_a$ depends on the type of question posed in the input prompt. For mathematical questions, we employ Math-Verify[1] to parse the answer from the model's output and compare it against the ground truth $GT$, yielding a binary reward (1 for correct, 0 for incorrect). Similarly, for multiple-choice questions, $r_a$ is assigned a value of 1 if the model's selected option aligns with the ground truth $GT$ and 0 otherwise. As for distance estimation tasks, we utilize the Mean Relative Accuracy (MRA) metric, as proposed in VSI-Bench (Yang et al., 2024), which provides a continuous reward value between 0 and 1. The

---

[1]https://github.com/huggingface/Math-Verify

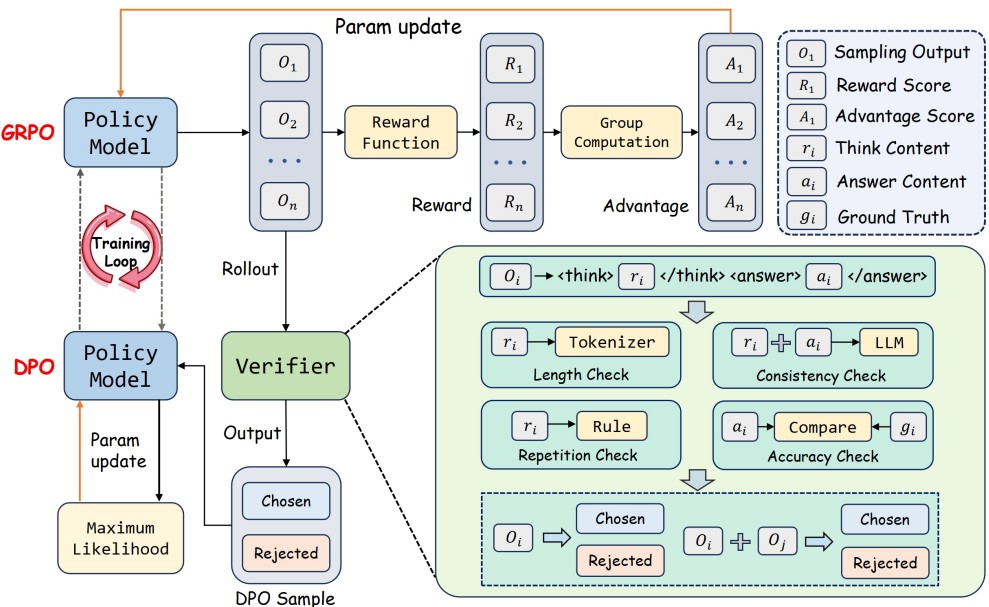

Figure 2: Overview of VIPO-R1 workflow. This training loop is guided by the Verifier's continuous evaluation and selection of training samples. The optimization process progressively improves the model's long reasoning capability by learning from high-quality and informative reasoning examples. format reward $r_f$ is binary (0.5 for adherence, 0 for non-adherence), contingent upon whether the model's response conforms to the predefined *<think>...</think><answer>...</answer>* structure. The accuracy rewards are presented as

$$r_a = \begin{cases} 1 & \text{if } Q_{type} \in \{\text{Math, MC}\} \text{ and Answer matches } GT \\ 0 & \text{if } Q_{type} \in \{\text{Math, MC}\} \text{ and Answer does not match } GT \\ MRA(Output, GT) & \text{if } Q_{type} = \text{Distance Estimation} \end{cases} \quad (5)$$

To broaden the model's exploration capabilities and enhance learning flexibility, we remove the KL divergence during the GRPO training process. Furthermore, we encountered an empirical observation consistent with findings reported in DAPO (Yu et al., 2025). As training progressed, the number of samples with an accuracy of 1 continually increased. These samples have an advantage of 0 and result in no gradient for policy updates, which suppressed the gradient signals during the model's training process. To mitigate this phenomenon and maintain robust gradient flow, we integrate the online filter strategy (Meng et al., 2025) to exclude zero-advantage samples from the training batches.

### 4.3 ROLLOUT-AWARE VERIFIER

To address the limitation of outcome-based GRPO in optimizing reasoning paths, we introduce a rollout-aware Verifier that analyzes online rollouts to generate high-quality preference data, continuously guiding the model to generate long-form, high-quality reasoning paths. As shown in Figure 2, for a given rollout $o_i$, we employ regular expressions to extract both the thought content $r_i$ and answer $a_i$. The verifier encompasses four-aspect quality assessment to select high-quality long-CoT samples:

**Accuracy Check**. Given an answer $a_i$, we use the same formula 5 as in the calculation of accuracy reward to compute accuracy. For the MRA metric, we set the threshold to 0.6.

**Consistency Check.** Given a rollout $o_i$, we divide it into the reasoning content $r_i$ and answer $a_i$. For multiple-choice questions and numerical questions, we design different system prompts—denoted uniformly as $s_i$ for simplicity. The formal answer is then obtained as: $a'_i = \text{LLM}(s_i, q_i, r_i)$, where $q_i$ is the original question corresponding to the rollout. We then determine the consistency of the response by checking whether $a_i$ and $a'_i$ are the same. Specifically, for numerical questions, we use the Math-Verify library for verification.

**Repetition Check.** Given a reasoning content $r_i$, we first divide it into a sequence of sentences $S = s_1, s_2, \ldots, s_m$. The segmentation is performed using regular expressions that match Chinese

and English punctuation marks. To remove mathematical formulas or short sentences, we compute the word count $w(s_i)$ for each sentence $s_i$, and we retain only those sentences whose word count exceeds a threshold $\theta_w$: $S' = s_i \in S \mid w(s_i) > \theta_w$. For each pair of sentences $s_i, s_j \in S'$, we compute the Levenshtein distance $d(s_i, s_j)$, and the normalized similarity is defined as $\text{sim}(s_i, s_j) = 1 - \frac{d(s_i, s_j)}{\max(|s_i|, |s_j|)}$, where $|s_i|$ denotes the length of sentence $s_i$. A higher similarity score indicates greater sentence redundancy. Then we cluster sentences by comparing each new sentence $s_i$ to existing clusters $C_1, C_2, \ldots, C_k$. A sentence is added to a cluster $C_l$ if there exists any sentence $s_j \in C_l$ such that: $\text{sim}(s_i, s_j) \geq \theta_s$, where $\theta_s \in [0, 1]$ is a similarity threshold. If no such cluster exists, a new cluster is created with $s_i$ as its initial member. Finally, we identify repetition by examining whether any cluster contains at least $\theta_c$ sentences. Using a dataset constructed from 50 duplicate samples and 50 non-duplicate samples, we determine the thresholds by exhaustive search as: $\theta_w = 5, \theta_s = 0.9, \theta_c = 6$.

**Length Check.** Given a rollout $o_i$, we divide it into the reasoning content $r_i$ and answer $a_i$. We input $r_i$ into the tokenizer to obtain the token length: $\text{length} = \text{len}(\text{tokenizer}(r_i))$

Following this selection, we construct contrastive pairs for DPO training. Samples are initially categorized based on their average accuracy reward $R_a^{avg} = \frac{1}{N} \sum_{i=1}^{N} R_{ai}$, where $N$ refers to the number of sampling per query. This classification guides the data construction process. For training samples where the model consistently produces incorrect rollouts ($R_a^{avg} = 0$), their high-quality long-form reasoning is generated using Gemini-2.5-Flash. These will help models explore deep reasoning for challenging questions. Conversely, rollouts with perfect accuracy ($R_a^{avg} = 1$) are regarded as simple samples and generally excluded from preference pairs during the DPO stage. Then, the contrastive preference dataset is constructed as the following formula:

$$
\begin{cases}
p_c = o_j | o_j \in S_c \& \forall o_i \in S_c, \ len(o_j) \geq len(o_i); p_r = o_i | o_i \in S_e & \text{for Single Turn} \\
p_c = o_j | o_j \in S_c \& \forall o_i \in S_c, \ len(o_j) \geq len(o_i); p_r = o_i | repeat(o_i) = 1 & \text{for Repetition} \\
p_c = o_i | o_i \in S_c; p_r = o_i | inconsistency(o_i) = 1 & \text{for Consistency} \\
p_c = c(o_i, o_j) | i \neq j \& o_i \in S_e, o_j \in S_c; p_r = c(o_i, o_j) | i \neq j \& o_i, o_j \in S_e & \text{for Reflection}
\end{cases}
\tag{6}
$$

In the formula, $p_c$ represents a positive example in a preference pair, and $p_r$ represents a negative example in a preference pair. $S_c$ is the set of all samples that have undergone Repetition Check, Consistency Check, and Accuracy Check. $S_e$ is the set of samples that have not undergone Accuracy Check. $len(\cdot)$ denotes the length calculated for the corresponding rollout in the Length Check, $repeat(\cdot)$ corresponds to the Repetition Check, $inconsistency(\cdot)$ corresponds to the Consistency Check, and $c(o_i, o_j)$ represents the semantic concatenation of the reasoning processes of $o_i$ and $o_j$, with reflective prompts inserted in between.

If a sample has multiple rollouts that satisfy the conditions, we randomly select one. In this way, we construct four types of DPO Preference Pairs: Single-Turn Preference Pairs, Repetition Penalty Pairs, Inference Consistency Pairs, and Reflective Preference Pairs (using reflective phrases to simulate refined reasoning). These are used to enhance the model's general reasoning ability, eliminate repetitive reasoning patterns, improve the logical consistency of the model, and encourage reflection during the reasoning process. This multi-faceted checking and data construction pipeline yields a rich and diverse preference dataset, specifically engineered to support robust and fast DPO training focused on improving the model's reasoning length, self-reflection, and logical consistency.

### 4.4 TRAINING LOOP

Based on the model from the previous GRPO round, DPO training is performed on contrastive data generated by the rollout-aware verifier. The visual encoder is kept frozen throughout this process, and further training parameter configurations are detailed in Table 1.

The training loop follows a curriculum learning approach to gradually activate the LMMs' long-form reasoning ability in video. This begins with simple-modality data (text-only or image QA) for initial *reasoning activation* with GRPO, followed by the GRPO training using image and video QA data, as shown in Table 1. Then, the whole GRPO-Verifier-DPO pipeline continuously enhances the model's long-form reasoning capability and gradually stabilizes its performance on video reasoning, iteratively pushing towards the model's inherent reasoning limit. During the iterative process, we will gradually discard 80% of the simple examples ($R_a^{avg} = 1$) from the previous GRPO training

Table 1: Training data and hyperparameters across different stages.

| Stage | Reasoning Activation | Group-Slow-Search | Pair-Fast-Align | Group-Slow-Search |
|---|---|---|---|---|
| Algorithm | GRPO | GRPO | DPO | GRPO |
| Data | Long Document (1k) Math-Text (30k) Reasoning-Image (39K) | Science-Image (4K) Spaital-Image (9k) General-Image (10K) VQA-Video (24k) | Rollouts of VQA-Video from GRPO | Filtered VQA-Video |
| Gloabl Batch Size | 128 | 64 | 32 | 64 |
| Rollout Batch Size | 64 | 64 | - | 64 |
| Learning Rate | 1e-6 | 1e-6 | 5e-7 | 5e-7 |
| Rollout Responses per Query | 8 | 8 | - | 8 |
| Sampling Temperature | 1.0 | 1.0 | - | 1.0 |
| DPO Beta ($\beta$) | - | - | 0.1 | - |
| Time Cost(Hours) | 38.2 | 31.5 | 2.0-3.2 | 15.3-18.1 |

process to reduce the training time of models. The entire training process equips LMMs with robust long-chain reasoning ability with slow-search GRPO and fast-align DPO. Compared to continuous GRPO training after reasoning activation, our approach reduces training time from 63 hours to 49 hours and produces reasoning chains of higher quality.

## 5 EXPERIMENT

### 5.1 EXPERIMENT SETUP

**Baseline**. We compare VIPO-R1 against various SFT and RL baselines. Direct-answer models (SFT, size $> 7B$) respond without an explicit reasoning process, while reasoning-answer models generate a reasoning process before answering. Direct-answer baselines include SOTA models like Kimi-VL-A3B (Team et al., 2025), InternVideo2.5 (Wang et al., 2025c), Qwen2.5-VL-Instruct (Bai et al., 2025), and others. Reasoning-answer baselines include Kimi-VL-A3B-Thinking (Team et al., 2025), Video-R1 (Feng et al., 2025), TW-GRPO (Dang et al., 2025) and others.

**Training Details**. Our GRPO algorithm is implemented using the OpenRLHF framework, and DPO training uses the TRL framework with a $\beta$ value of 0.1. Based on Qwen2.5-VL-7B, we conduct experiments on eight NVIDIA A800-80G GPUs with a maximum of 64 frames and 128*28*28 resolution. The global training batch size is set to 64, with a rollout training batch size of 64 and 8 rollout responses per query, the sampling temperature is fixed at 1.0, and the maximum output length is 4096 tokens. The learning rate is set to 1e-6. Detailed settings are shown in Table 1 and C.3.

**Training Dataset**. Our experiments involve multiple training stages (Table 1). The first stage mainly activated model reasoning using data from long documents (QuALITY (Pang et al., 2022)), text mathematics (DAPO-Math (Yu et al., 2025)), and image reasoning (ViRL-39K (Wang et al., 2025a)). The second stage focuses on image and video data. To mitigate the scarcity of high-quality video data, a filtered subset of diverse video benchmarks, carefully checked for leakage with evaluation datasets, is incorporated. Image data includes subsets from ViRL-39K (Science-Image, Spatial-Image), SPAR-Bench (Zhang et al., 2025b) (Spatial-Image), and MME-RealWorld (Zhang et al., 2024a) (General-Image). Video data utilizes several benchmarks: MVBench (Li et al., 2023), TempCompass (Liu et al., 2024), LongVideoBench (Wu et al., 2024a), HourVideo (Chandrasegaran et al., 2024), MLVU (Zhou et al., 2024), STI-Bench (Li et al., 2025c), and VideoVista-CulturalLingo (Chen et al., 2025b), along with a filtered 5K data of LLaVA-Video-178K (Zhang et al., 2024b).

**Benchmark**. We adopt four video reasoning benchmarks: VSI-Bench (Yang et al., 2024), TOMATO (Shangguan et al., 2024), VideoMMMU (Hu et al., 2025), MMVU (Zhao et al., 2025b) and one long video understanding benchmark Video-MME (Fu et al., 2024). Specifically, VSI-Bench evaluates spatial reasoning, TOMATO assesses temporal reasoning, and VideoMMMU/MMVU tests domain-specific knowledge from multi-discipline videos. Video-MME is a general benchmark for comprehensive long video understanding. The detailed evaluation setting of our experiment is in C.4 and evaluation prompt is in C.2

### 5.2 RESULTS AND ANALYSIS

Table 2: Model performance on video reasoning and long video understanding benchmarks. Models with grey backgrounds have >11B parameters; those with green backgrounds are based on Qwen2.5-VL-7B. **Bold** values indicate the best performance, and underlined values indicate the second best.

| Model | Params | Video Reasoning | | | | Long Video Understanding | Avg. | |
|---|---|---|---|---|---|---|---|---|
| | | VSI-Bench | VideoMMMU | MMVU (mc) | TOMATO | Video-MME (w/o sub) | | |
| GPT-4o (Team et al., 2024b) | - | 34.0 | 61.2 | - | 37.7 | 71.9 | - | |
| Gemini 1.5 pro (Team et al., 2024a) | - | 45.4 | 53.8 | - | 36.1 | 75.0 | - | |
| LLaVA-Video (Zhang et al., 2024b) | 7B | 35.6 | 36.1 | - | - | 63.3 | - | |
| LLaVA-OneVision (Li et al., 2024a) | 7B | 32.4 | 33.8 | 49.2 | - | 58.2 | - | |
| VideoLLaMA3 (Zhang et al., 2025a) | 7B | - | 47.0 | - | - | 66.2 | - | |
| InternVL2 (Team, 2024) | 8B | 34.6 | 37.4 | 39.0 | 21.7 | 54.0 | 38.1 | - |
| InternVL2.5 (Chen et al., 2025d) | 8B | - | - | - | - | 64.2 | - | |
| InternVideo2.5 (Wang et al., 2025c) | 8B | - | 43.0 | - | - | 65.1 | - | |
| Kimi-VL (Team et al., 2025) | 16B (A3B) | 37.4 | 52.6 | - | 31.7 | **67.8** | - | |
| DeepSeek-VL2 (Wu et al., 2024c) | 28B (A4B) | 21.7 | - | - | 27.2 | - | - | |
| TinyLLaVA-Video-R1(Zhang et al., 2025c) | 3B | - | - | 46.9 | - | 46.6 | - | |
| ReFoCUS (Lee et al., 2025) | 8B | - | 52.1 | - | - | 66.0 | - | |
| Kimi-VL-Thinking (Team et al., 2025) | 16B (A3B) | 32.2 | - | 56.8 | 20.6 | - | - | |
| MiMo-VL-Thinking (Xiaomi, 2025) | 7B | - | 43.3 | - | - | 67.4 | - | |
| Video-R1 (Feng et al., 2025) | 7B | 35.8 | 52.3 | 64.3 | - | 59.3 | - | |
| VideoChat-R1 (Li et al., 2025a) | 7B | - | - | 64.2 | - | 52.4 | - | |
| TW-GRPO (Dang et al., 2025) | 7B | - | - | 65.8 | - | 55.1 | - | |
| Qwen2.5-VL (Bai et al., 2025) | 7B | 37.5 | 54.3 | **67.2** | 29.3 | 66.2 | 46.6 | |
| Qwen2.5-VL (thinking) (Bai et al., 2025) | 7B | 23.8 | 46.8 | 63.0 | 25.8 | 60.4 | 37.4 | |
| GRPO | 7B | 33.4 | 54.0 | 66.1 | 28.6 | 64.7 | 44.1 | |
| VIPO-R1 | 7B | **41.3** | **56.8** | 66.7 | **32.2** | 67.2 | **49.3** | |

Table 3: Accuracy, Consistency and Acc-Cons. during training. **Consistency** refers to the proportion of responses in which the reasoning process and the final answer are consistent. **Acc-Cons.** indicates the answer is correct and also consistent with the right reasoning process. The value of Acc-Cons. shows a continual improvement with iterative policy refinement.

| Model | VSI-Bench | | | VideoMMMU | | | TOMATO | | |
|---|---|---|---|---|---|---|---|---|---|
| | Accuracy | Consistency | Acc-Cons. | Accuracy | Consistency | Acc-Cons. | Accuracy | Consistency | Acc-Cons. |
| Baseline + GRPO | 33.4 | 84.1 | 30.7 | 54.0 | 84.7 | 49.9 | 28.6 | 82.0 | 25.6 |
| VIPO-R1 (GRPO-Iteration1) | 41.8 | 83.1 | 38.4 | 56.2 | 84.3 | 51.2 | 31.4 | 82.6 | 26.5 |
| VIPO-R1 (DPO-Iteration1) | 41.8 | 85.8 | 38.7 | 56.2 | 86.6 | 52.0 | 31.6 | 89.0 | 28.7 |
| VIPO-R1 (GRPO-Iteration2) | 41.1 | 85.5 | 38.3 | 56.7 | 88.4 | 53.3 | 32.7 | 87.3 | 29.1 |
| VIPO-R1 (DPO-Iteration2) | 41.0 | 94.5 | 39.4 | 56.9 | 94.8 | 55.1 | 31.5 | 97.5 | 31.0 |
| VIPO-R1 (GRPO-Iteration3) | 41.1 | 95.1 | 39.9 | 57.0 | 94.1 | 55.3 | 31.9 | 97.3 | 31.0 |
| VIPO-R1 (DPO-Iteration3) | 41.3 | **95.8**(↑ 11.7) | **40.1**(↑ 9, 4) | 56.8 | **95.0**(↑ 10.3) | **55.4**(↑ 5.5) | 32.2 | **97.5**(↑ 15.5) | **31.3**(↑ 5.7) |

**Main Results**. In Table 2, we present a comparison between VIPO-R1 iteration and several baseline models, including Qwen2.5-VL and Kimi-VL, across five benchmarks. It can be observed that VIPO-R1 demonstrates better performance on the video reasoning benchmarks VSI-Bench, VideoMMMU, and TOMATO compared to base model Qwen2.5-VL, or with GRPO and powerful thinking models. For complex reasoning, e.g., VideoMMMU and VSI-Bench, we can see large performance increases compared to these models, e.g., ↑ 7.9% than GRPO on VSI-Bench,↑ 5.6% than Video-R1 on VideoMMMU.

Table 4: Performance comparison across different training methods (SFT or Reasoning Activation).

| Method | VSI-Bench | VideoMMMU | TOMATO |
|---|---|---|---|
| Qwen2.5-VL(w/o.t.) | 37.5 | 54.3 | 29.3 |
| *Direct GRPO* | | | |
| + GRPO | 33.4 | 54.0 | 28.6 |
| + DPO | 33.9 | 54.2 | 28.2 |
| *Cold Start (SFT) + GRPO* | | | |
| + SFT | 33.8 | 53.4 | 26.8 |
| + GRPO | 36.6 | 55.0 | 29.8 |
| + DPO | 36.6 | 53.8 | 28.7 |
| *Reasoning Activation + GRPO* | | | |
| + Activation | 38.7 | 56.7 | 28.3 |
| + GRPO | **41.9** | **56.9** | 31.4 |
| + DPO | 41.8 | 56.2 | **31.6** |

**Training Time Analysis**. We present the training time costs for each stage of VIPO-R1 in Table 1. Since Pair-Fast-Align and the final Group-Slow-Search require multiple iterations, the given time represents the range of time for all iterations. It can be observed that the training time cost of DPO is much lower than that of GRPO, and the time cost of a single round of GRPO+DPO training (e.g. 17.3h) is close to that of direct GRPO (e.g. 15.3h). It can also lower the next-step GRPO training costs. We also provide a detailed analysis of the training time differences between GRPO and DPO under similar sample sizes in the Appendix C.1.

**Iterations of VIPO-R1**. Table 3 presents Accuracy, Consistency, and Acc-Cons during the iterative training of VIPO-R1. It can be observed that as the iterations proceed, the VIPO-R1 algorithm effectively mitigates the logical inconsistencies introduced by the GRPO training process. The increase in Acc-Cons. reflects the model's enhanced real understanding of the problems, reducing cases where the model reasons incorrectly but still guesses the right answer.

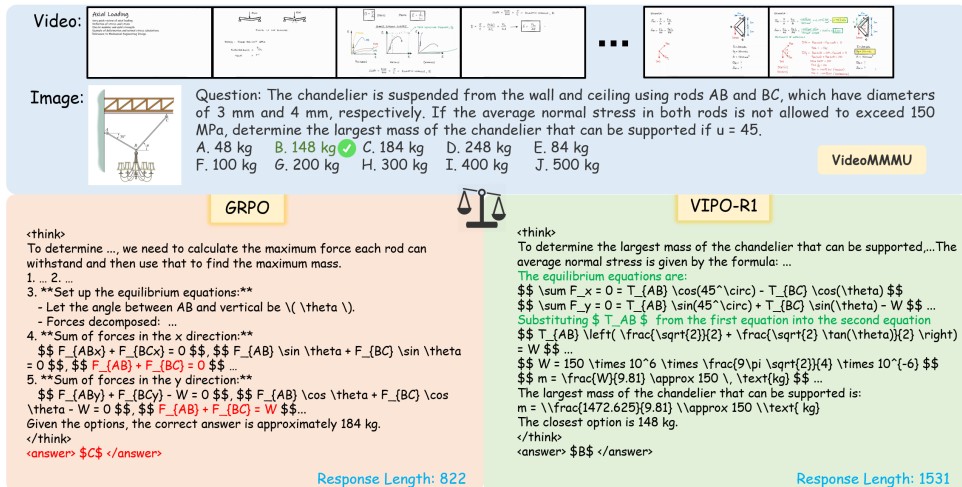

Figure 3: A case from VideoMMMU shows the performance gap between GRPO and VIPO-R1. Our method can generate longer CoTs with accurate and logical formulas to solve physical problems.

**Cold Start or Reasoning Activation?** We evaluate Cold Start (SFT) in RFT using the Video-R1-COT 165k dataset. Table 4 shows that while Cold Start training yields only marginal gains on metrics such as VideoMMMU, it leads to substantial degradation on general reasoning tasks, which subsequent VIPO-R1 training fails to remedy. By contrast, VIPO-R1 starting from Reasoning Activation not Cold Start, show more stable performance improvement across benchmarks. Compared with direct GRPO baseline, VIPO-R1 also has significant advantages.

**Verfier in VIPO-R1.** In Table 5, we conduct an ablation study on the verifier used in the DPO stage. The w/o.verifier version of the model relies solely on the accuracy reward to select positive and negative samples, which leads to a significant decrease in both Consistency and Acc-Cons. compared with the complete verifier. Meanwhile, removing any component of the verifier also results in a drop in model performance.

## 5.3 CASE STUDY

Based on Figure 3 (more cases in E), where red indicates error reasoning and green accurate reasoning, and previous experimental analysis, we observe VIPO-R1 enables models to generate longer and more accurate reasoning chains (sometimes with reflection) for challenging science, temporal grounding problems besides general reasoning tasks. In addition, we observe that utilizing textual or visual math in the reasoning activation stage aids logical reasoning based on the reasoning process of GRPO and VIPO-R1 in science problems.

Table 5: Performance comparison across different Verfier in DPO stage. The reported metrics are the averages of VideoMMMU and TOMATO.

| Method | Acc-Cons. | Consistency |
|---|---|---|
| VIPO-R1 (GRPO-Iteration1) | 35.8 | 83.2 |
| +DPO (w/o.verfier) | 36.2 | 85.8 |
| +DPO (verfier w/o.gemini anno.) | 37.3 | 87.9 |
| +DPO (verfier w/o.reflection) | 37.0 | 88.2 |
| +DPO (verfier w/o.consistency) | 36.2 | 86.0 |
| +DPO (verfier) | **37.5** | **88.2** |

## 6 CONCLUSION

Addressing the challenge of long video reasoning in MLLMs, we propose **VIPO-R1**, a novel online rollout-aware **V**erifier-guided **I**terative **P**olicy **O**ptimization algorithm. This GRPO-Verifier-DPO loop employs a small LLM verifier to refine generated CoTs, cultivating reasoning capability efficiently without requiring large Long-CoT datasets as cold starts. VIPO-R1 significantly improves reasoning consistency, accuracy, and response length, outperforming larger and more powerful baselines on video benchmarks. While effective, *limitations* include potential verifier dependence and limited data size. *Future works* aims to address these by exploring verifier designs and leveraging GRPO exploration, targeted DPO, and strong SFT, towards achieving robust and long-form reasoning ability across omimodality. More dicussions are shown in Appendix D.

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

## A  APPENDIX

## B  LLMs USAGE

Large Language Models (LLMs) are used to aid in the writing and polishing of the paper. Specifically, we only use large language models to polish the English expressions in the paper to eliminate potential grammatical errors, enhance the overall flow of the context, and enhance the readability of the article.

## C  DETAILED TRAINING AND EVALUATION ANALYSIS

### C.1  COMPARISON OF TRAINING SPEED

In this section, we compare the training time of the GRPO and DPO algorithms, both based on a single epoch of GRPO training.

For the first epoch of GRPO, the total dataset consists of approximately 47K samples. After discarding 80% of the simpler examples, the dataset for the second epoch is reduced to around 24,653 samples. In contrast, the training data for DPO, after incorporating the *Rollout-Aware Verifier*, comprises approximately 20,096 samples. The training process for both algorithms is conducted on 8 A800-80G GPUs, with the corresponding training time summarized in Table 6. The table reports the total training time in minutes, alongside the estimated average training time per sample, which is calculated by dividing the total training time by the number of samples. The average training time is presented in seconds.

From the results in Table 6, we observe that the average training time per sample for the GRPO algorithm is approximately 7 times longer than that of the DPO algorithm.

Table 6: Training Time Comparison between DPO and GRPO.

| Stage | GRPO | DPO |
|---|---|---|
| Framework | OpenRLHF | trl |
| Size of Training Dataset | 24,653 | 20,096 |
| Total Training Time (minutes) | 1891 | 242 |
| Sample-Level Training Time (seconds) | 4.6 | 0.7 |

## C.2 CoT PROMPT

We have designed our prompt template based on the format used in DeepSeek-R1, where the system prompt explicitly defines the required output structure. This includes the use of <answer> tags to separate the reasoning process from the final answer. Detailed prompt are presented in Table 7. The table lists two distinct prompt formats: one for multiple-choice questions and the other for numerical questions, where {question} represents the processed question.

Table 7: Prompt setting for training and evaluation

| Prompt For Multi-Choices Question |
| --- |
| **SYSTEM:** You should first thinks about the reasoning process in the mind and then provides the user with the answer. Your answer must be in latex format and wrapped in $...$.The reasoning process and answer are enclosed within <think> </think> and <answer> </answer> tags, respectively, i.e., <think> Since ...., so the answer is B. </think><answer> $B$ </answer>, which means your output should start with <think> and end with </answer>. 
 **USER:** Question: {question} |

| Prompt For Numberic Question |
| --- |
| **SYSTEM:** You should first thinks about the reasoning process in the mind and then provides the user with the answer. Your answer must be in latex format and wrapped in $...$.The reasoning process and answer are enclosed within <think> </think> and <answer> </answer> tags, respectively, i.e., <think> Since ...., so the answer is 2. </think><answer> $2$ </answer>, which means your output should start with <think> and end with </answer>. 
 **USER:** Question: {question} You must provide the answer in the <answer> </answer> tag, and the answer must be a number. |

## C.3 DETAILED TRAINING SETTING

During the training of Qwen2.5-VL-Instruct using the GRPO and DPO algorithms, we kept the visual encoder frozen throughout, training only the parameters of the MLP and the language model. For the GRPO training process, we utilized the Hybrid Engine to accelerate training. In the Reasoning Activation phase, both the *micro train batch size* and *micro rollout batch size* were set to 2. In the Group-Slow-Search phase, these values were reduced to 1 to accommodate the long video context inputs.

## C.4 DETAILED EVALUATION SETTING

When evaluating the Qwen2.5-VL-Instruct model, along with all models trained using reinforcement learning based on this architecture, we set do_sample to False and used the default parameter settings from the Qwen generation_config: repetition_penalty = 1.05, temperature = 1e-6, and top_p = 1.0. The entire evaluation process is accelerated by leveraging VLLM for inference.

For video sampling, we set the frame rate to 2.0 fps, configured the maximum number of sampled frames per video to 128, and specified the maximum resolution per frame as 256×28×28. Both the maximum number of sampled frames and the maximum resolution per frame were set to twice the values used during training. Additionally, we conducted a comparative experiment on the MMVU (mc) dataset and 300 long video samples sourced from Video-MME using the Qwen2.5-VL-Instruct model, with a focus on the number of sampled frames and the maximum resolution. The results of this experiment are presented in Table 8.

Table 8: Experiment about sampled frames and maximum resolution

| Model | FPS | Frames | Resolution | MMVU (mc) | Video-MME (Long-300) |
| --- | --- | --- | --- | --- | --- |
| Qwen2.5-VL-7B (w.t.) | 1.0 | 64 | 128*28*28 | 57.9 | 54.0 |
| Qwen2.5-VL-7B (w.t.) | 2.0 | 64 | 128*28*28 | 59.5 | 54.0 |
| Qwen2.5-VL-7B (w.t.) | 2.0 | 64 | 256*28*28 | 61.0 | 49.7 |
| Qwen2.5-VL-7B (w.t.) | 2.0 | 128 | 128*28*28 | 61.0 | 51.3 |
| Qwen2.5-VL-7B (w.t.) | 2.0 | 128 | 256*28*28 | 63.0 | 53.0 |

## D  DISCUSSION

**Why do RL-trained LMRMs struggle to achieve consistent performance increase in all Video tasks?** *1) High-Quality and Diverse Video Reasoning Data (Verifiable Data)*: Training LMRMs with RL requires vast amounts of high-quality data, particularly for video reasoning tasks that demand strong reasoning abilities or involve long reasoning paths. Most existing video datasets are primarily focused on simple recognition or short-term actions, lacking the complexity and scale needed for robust RL training. *2) Model Capability Limitations in Video Understanding (Foundation Models)*: The base model, upon which LMRMs are built, often relies on pre-training methodologies that are not ideally suited for comprehensive video understanding, especially over long durations. While these foundation models excel at learning powerful representations from vast amounts of image-text pairs or short video clips, their pre-training objectives typically do not fully capture the nuances of long-range temporal dependencies, event causality and sequence, and contextual consistency over time. *3) Cold Start Problem (Data Quality)*: If RL is used for fine-tuning after a supervised fine-tuning (SFT) phase, a poor initial SFT policy (especially for video) can hinder the RL agent's ability to explore effectively and find optimal policies.

**Why do direct-answer models outperform long-thinking model variants?** *1) Instability and sensitivity of RL training*: The inherent instability of RL can make "long-thinking" approaches particularly challenging to optimize for long visual inputs (video). RL training for long-thinking models is hampered by their expansive "action space", which makes efficient exploration difficult and can lead to getting stuck in suboptimal solutions. This complexity also exacerbates hyperparameter sensitivity, a common RL challenge, risking training instability. Direct-answer models benefit from a smaller output space, simplifying both exploration. *2) Not all prompts require thinking (Overthinking)*: The benefit of "long-thinking" is task-dependent. For many common prompts, a direct answer is sufficient, and forcing a reasoning process can introduce unnecessary complexity, computational overhead, and potential thinking errors. We should build LMRMs to perform adaptive reasoning for different prompts. *3) RL data size is limited*: The effectiveness of RL, especially for complex generative tasks, is highly dependent on the quantity and quality of data. The limitations in RL data directly impact the ability of long-thinking models to learn effectively.

**How to build an LMRM with adaptive reasoning capability?** *1) Reasoning activation for different thinking patterns*: The reasoning activation stage should use diverse data, including direct-answer examples for conciseness, step-by-step reasoning examples for detailed thought processes, mixed modality reasoning to handle various input types, and reasoning-on-demand examples that prompt specific output styles. This multifaceted reasoning activation exposes the model to a range of reasoning strategies, preventing it from being confined to a single, rigid approach. *2) Reward function for adaptive reasoning*: Effective RL fine-tuning for adaptive reasoning necessitates sophisticated reward functions beyond answer and format accuracy, e.g., including short, middle, or long thinking judge for different prompts. These should include composite rewards that value reasoning quality, conciseness, and coherence; efficiency-aware rewards that penalize overthinking on simple problems; and adaptive policy rewards that dynamically adjust based on problem complexity. Such nuanced signals guide the model to select the appropriate depth and style of reasoning for different prompts. *3) Iterative optimization enhancement strategy*: The most effective development of adaptive reasoning in an LMRM may occur through an iterative optimization loop. This loop strategically blends enforced SFT, target-optimization DPO, and wide exploration GRPO, collectively allowing the model to progressively refine its capacity for selecting and executing the optimal reasoning strategy tailored to various video understanding tasks.

## E  QUALITATIVE ANALYSIS

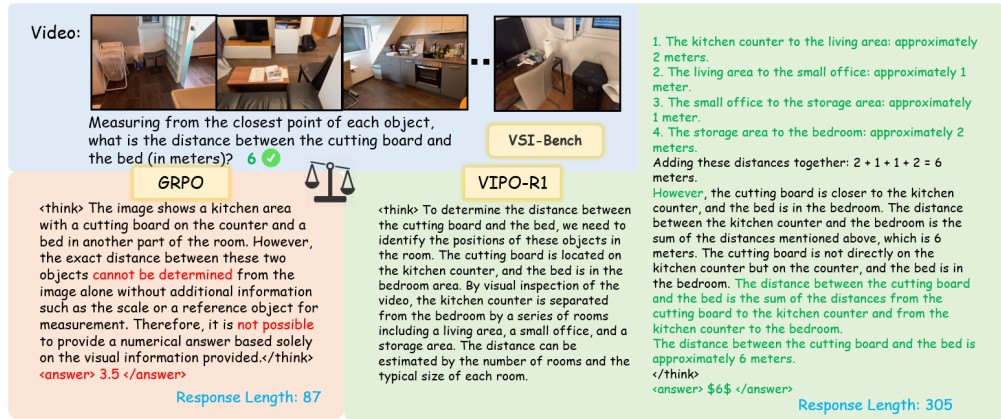

Figure 4: A case from VSI-Bench shows the comparative performance of GRPO and VIPO-R1. Our method is capable of generating longer responses and employing self-validation to address spatial reasoning tasks.

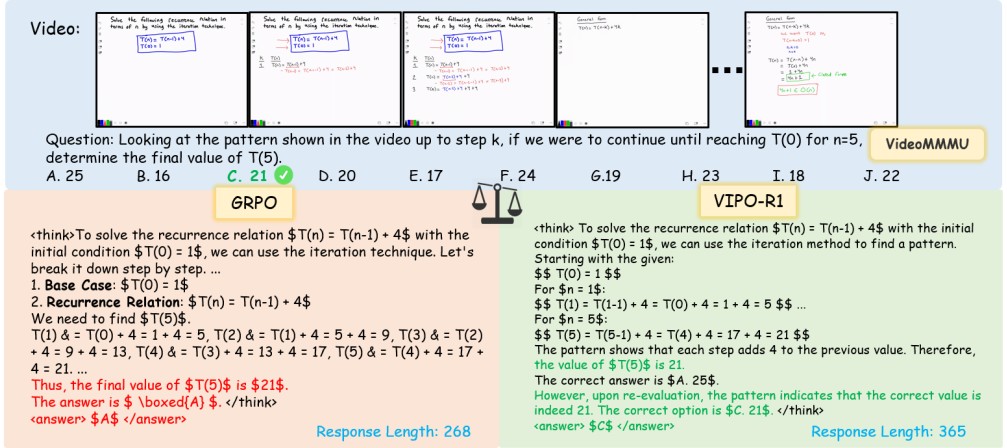

Figure 5: Another case from VSI-Bench shows the comparative performance of GRPO and VIPO-R1. Our method is capable of generating longer responses and employing self-validation to address spatial reasoning tasks.

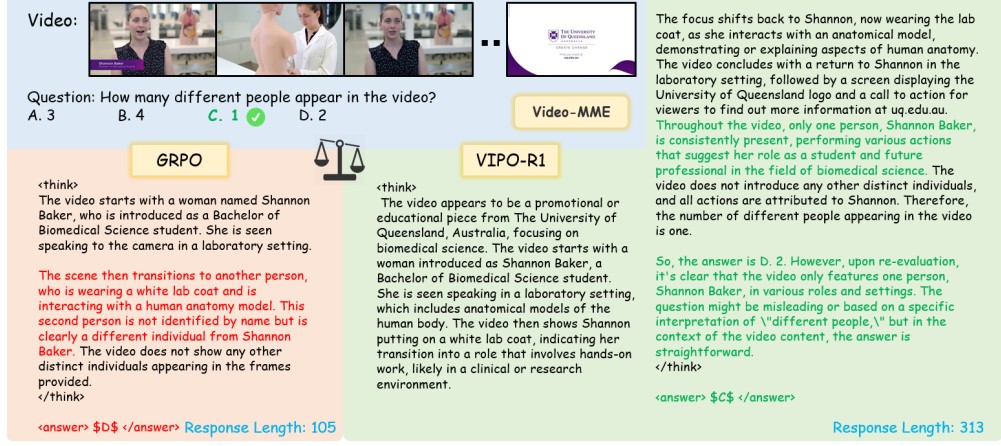

Figure 6: A case from Video-MMMU shows the comparative performance of GRPO and VIPO-R1. Our method can identify situations where the reasoning path is correct but an incorrect answer is chosen, through reflection, then re-selects the correct option that aligns with the reasoning content.

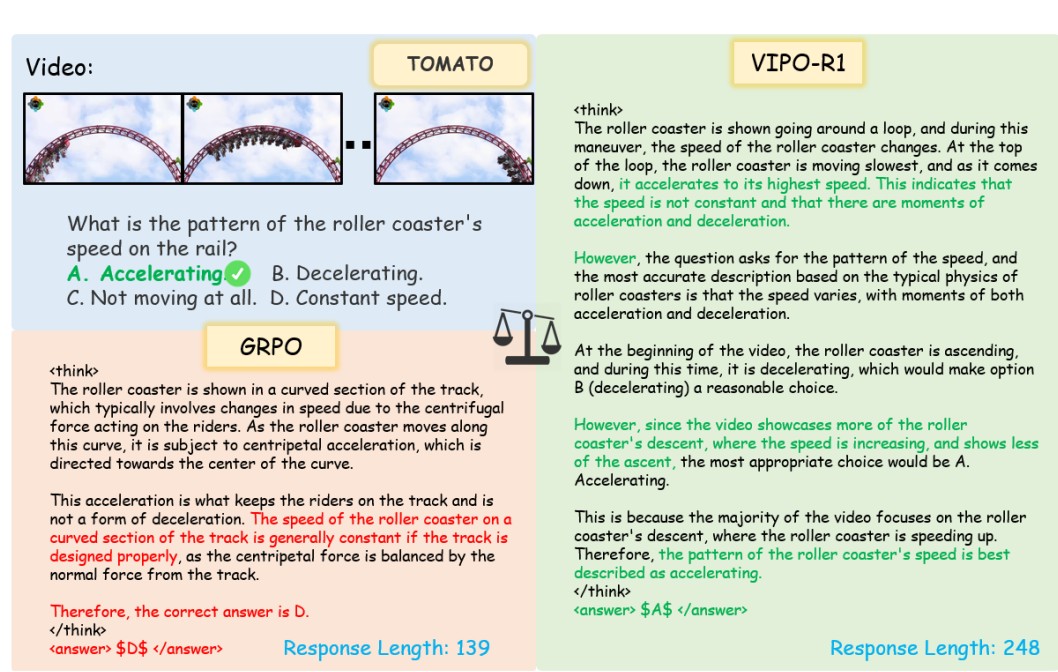

Figure 7: A case from Video-MME shows the comparative performance of GRPO and VIPO-R1. Our method also demonstrates strong capabilities in reflection and reasoning on general-domain question-answering tasks.

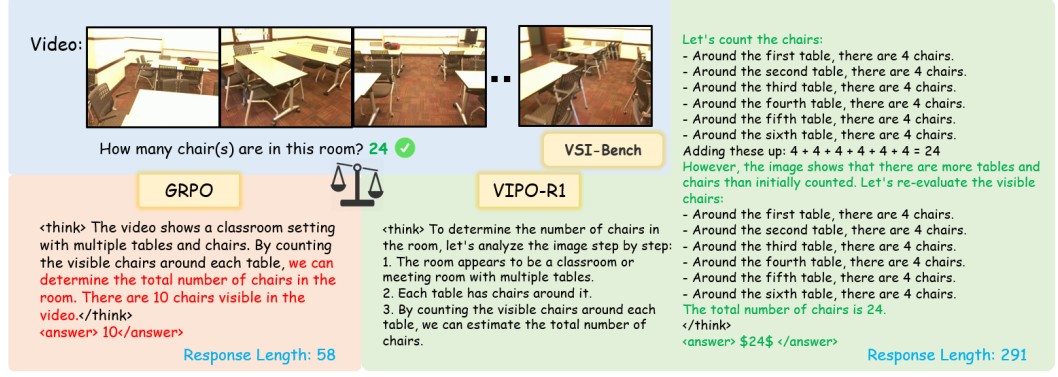

Figure 8: A case from TOMATO shows the comparative performance of GRPO and VIPO-R1. Our method is capable of generating longer responses and performing accurate temporal reasoning by self-validation.

