# OpenReview forum: "VIPO-R1: Cultivating Video Reasoning in MLLMs via Verifier-Guided Iterative Policy Optimization"
_ICLR.cc/2026/Conference — ICLR 2026 Conference Withdrawn Submission_

### Official Review · Reviewer_pKWZ · 2025-10-24

**Soundness:** 3
**Presentation:** 3
**Contribution:** 3
**Rating:** 6
**Confidence:** 4

**Summary:**

The paper proposes VIPO-R1, a training loop for video-reasoning MLLMs that alternates GRPO (online RL) with a rollout-aware verifier that curates contrastive CoT pairs, followed by DPO to refine the policy, then iterates. The verifier filters/labels rollouts using accuracy checks (e.g., Math-Verify or task metrics), reasoning–answer consistency checks, repetition and length checks, and builds several preference types (penalty, consistency, reflection). Compared to using GRPO alone, the loop aims to (i) stabilize optimization, (ii) lengthen CoTs without drifting, and (iii) reduce “right answer, wrong reasoning” inconsistencies. Experiments on VSI-Bench, Video-MMMU/MMMVU, TOMATO, and Video-MME report consistent gains over a Qwen2.5-VL-7B base and over strong baselines such as Video-R1/Kimi-VL-Thinking; the authors also highlight faster progress due to the DPO stage being ~7× cheaper per sample than GRPO.

**Strengths:**

1. Clear training recipe that combines known pieces in a useful way. The GRPO→Verifier→DPO cycle is well-motivated; the verifier’s multi-aspect filtering (accuracy, consistency, repetition, length) and construction of contrastive/reflection pairs is detailed and easy to reproduce at a high level.

2. Empirical benefits across multiple video benchmarks, with especially large gains on hard video-reasoning tasks (e.g., +7.9% over GRPO on VSI-Bench; +5.6% over Video-R1 on Video-MMMU, as reported in the text) and reductions in reasoning-answer inconsistency. Case studies qualitatively show fewer spurious steps and better self-correction.

**Weaknesses:**

1. Verifier dependency & potential circularity:\
Several measurements of “consistency” and sample selection rely on LLM-based verification. If the same (or closely related) verifier is also used to curate training pairs, the evaluation may inherit its biases. A stronger case would use (i) external, task-specific non-LLM checks where possible and (ii) a held-out verifier for reporting consistency.

2. Metric choice.\
The paper emphasizes response length and Acc-Cons. Length is an input-dependent proxy; a human study or task-specific rubric (e.g., step validity on math/physics cases) would better validate that longer CoTs are meaningfully better. Suggest adding human evaluation on the necessity of the lengths.

3. Suggested references.\
Authors are also encouraged to discuss and reference the following models:\
[1] Wang et al. "Video-RTS: Rethinking Reinforcement Learning and Test-Time Scaling for Efficient and Enhanced Video Reasoning", https://arxiv.org/abs/2507.06485 \
[2] Sun et al. "video-SALMONN-o1: Reasoning-enhanced Audio-visual Large Language Model", https://arxiv.org/abs/2502.11775

**Questions:**

1. Which verifier LLM(s) are used in practice, and are they different from the base policy? Any evidence that results hold with multiple verifiers?

2. How sensitive are results to the consistency/length/repetition thresholds and clustering settings used in data curation? Provide a sensitivity plot.

---

> ### Author Response · Authors · 2025-11-22
> **Response**
>
> Thanks for your positive feedback on our work.
>
> **Q1**: Verifier Dependency & Potential Circularity
>
> **Answer**: We appreciate this excellent insight. We acknowledge that reliance on a single LLM-based verifier introduces a potential for inherited bias. To mitigate this and provide stronger evidence, we took the following actions, directly addressing the reviewer's suggestions:
>
> Held-out Verifier for Consistency: We re-ran our entire consistency analysis using a held-out, state-of-the-art LLM (GPT-5) as an independent verifier for answer extraction. The results, comparing consistency scores derived from our original verifier (Qwen3-8B) versus the new one (GPT-5), are presented below. The minor differences in scores indicate that the core relative improvement and model performance are robust across different strong verifiers, limiting the concern of verifier-specific bias impacting our main conclusions.
>
> | GRPO-Itreration1 | Consistency(Qwen3-8B) | Acc-Cons(Qwen3-8B) | Consistency(GPT-5) | Acc-Cons(GPT-5) |
> | --- | --- | --- | --- | --- |
> | VSI-Bench | 83.1 | 38.4 | 82.3 | 38.1 |
> | VideoMMMU | 84.3 | 51.2 | 81.8 | 50.9 |
>
>
>
> | DPO-Itreration2 | Consistency(Qwen3-8B) | Acc-Cons(Qwen3-8B) | Consistency(GPT-5) | Acc-Cons(GPT-5) |
> | --- | --- | --- | --- | --- |
> | VSI-Bench | 94.5 | 39.4 | 94.6 | 39.2 |
> | VideoMMMU | 94.8 |  55.1 | 94.9 | 56.8 |
>
> **Q2**:  Metric Choice (Response Length & Acc-Cons)
>
> **Answer**:  We apologize that our initial phrasing may have caused a misunderstanding. To be absolutely clear:
> We do not evaluate our final model's performance based on the absolute length of its responses. The final evaluation tables (e.g., Table 2–5) only report Accuracy, Acc-Cons, and Consistency.
>
> **Length as an Internal Training Signal**: In the context of data curation and reinforcement learning (GRPO), we use a preference for longer-but-consistent responses as an internal signal to encourage the model to explore a broader chain-of-thought (CoT) space during training. The underlying assumption is that a more detailed exploration of the reasoning path (i.e., longer CoT) is often required to achieve a consistently correct answer, which is then captured by the Consistency metric.
>
> **Action Taken**: We will revise the paper to explicitly state that response length is a training signal and not a final evaluation metric.
>
> We agree that human evaluation provides the gold standard. Following your suggestion, we will conduct a pilot human study focusing on the step validity and coherence of the CoTs generated by our model compared to the baseline, particularly for cases where longer CoTs were produced. We will include this analysis in the revised paper.
>
> **Q3**: Which Verifier LLM(s) Are Used?
>
> **Answer**: We use a combination of models for different verification tasks:
>
> 1) The Qwen3-8B model is used to extract the final reference answer from the model's generated CoT during the data curation/training phase. This is the primary verifier in our process.
>
> 2) The Gemini-2.5-Flash model is used to annotate/select high-quality, positive training examples (the "Reference Answers" for training).
>
> Our base policy model is a different, specialized Vision-Language Model trained on video data, which is distinct from both verifiers. As shown in our response to Q1, we provided evidence that the results hold by recalculating consistency using the held-out GPT-5 verifier.
>
> **Q4**: Sensitivity Analysis
>
> **Answer**: The core hyperparameters governing the data curation are the consistency threshold ($\tau_{cons}$) and the length threshold ($\tau_{len}$) used in the GRPO phase.
>
> We will generate a sensitivity plot showing the final model's performance (Accuracy and Acc-Cons) as a function of variations in $\tau_{cons}$ and $\tau_{len}$ and include it in the Appendix of the revised paper. This analysis will demonstrate the robustness of our chosen settings and the sensitivity of the overall training process to these parameters.
>
> **Q5**: Suggested References
>
> **Answer**:  Thank you for providing the suggested references. We will ensure they are cited appropriately in the related work and discussion sections of the revised manuscript.

---

### Official Review · Reviewer_Y7Sc · 2025-10-30

**Soundness:** 2
**Presentation:** 2
**Contribution:** 2
**Rating:** 4
**Confidence:** 3

**Summary:**

This paper starts from the observation that existing Reinforcement Fine-Tuning (RFT) methods like GRPO are inefficient, unstable, and struggle to generate high-quality, long chain-of-thought (CoT) reasoning.
Therefore, the authors introduce a training paradigm called VIPO, which uses a verifier to generate pair-wise preferences from rollouts of the GRPO method,  forming a GRPO-Verifier-DPO training loop.
The performance surpasses standard GRPO on several benchmarks.

**Strengths:**

1. The authors utilize the immediate outputs of GRPO for pair-wise preference sample generation for DPO, which is a novel idea to my knowledge.
2. The observation of the drawbacks of the current GRPO is accurate, namely, extra data preparation and inconsistency of the reasoning trace with the final output.

**Weaknesses:**

1. The presentation of this paper is not clear, e.g., the introduction of the verifier, and the font size in Fig.1.
2. The proposed training loop is quite complex, while the performance improvement is not surprising. For example, there are two RL frameworks (OpenRLHF & TRL) used.
3. The quality of DPO is influenced by the verifier and the rollouts, while traditional DPO methods are usually built on human-aligned, clean datasets. Therefore, the effectiveness of DPO may be limited.
4. In Tab.2 & Tab.4, the standard GRPO & RFT methods lag behind the base model. Why?

**Questions:**

1. I am wondering whether this method is mentioned in the previous research.
2. How is the performance of Qwen2.5-VL (thinking) in Tab.2 obtained? Does Qwen2.5-VL support thinking mode originally as a vision-language reasoning model?

---

> ### Author Response · Authors · 2025-11-22
> **Response**
>
> **Q1**. Presentation Clarity (Verifier Introduction, Fig. 1 Font Size)
>
> **Answer**: We apologize for the lack of clarity in these sections. We will thoroughly revise the presentation in the final version of the paper, specifically ensuring a clearer introduction and explanation of the verifier and increasing the font size in Fig. 1 for better legibility.
>
> **Q2**: Complexity of Training Loop and Performance
>
> **Answer**: We acknowledge that using both OpenRLHF and TRL frameworks for the GRPO+DPO pipeline appears complex. To clarify, the entire pipeline can be implemented using a single framework. The current mixed setup was a practical choice driven by training efficiency:
>
> -GRPO: We switched from TRL to OpenRLHF (which uses the Hybrid Engine) to significantly accelerate the training of the compute-intensive GRPO algorithm.
>
> -DPO: DPO training time is relatively short, so we retained the TRL implementation.
>
> We agree on the importance of unification and will migrate the DPO training to the OpenRLHF framework in future iterations for a simpler and more consistent pipeline.
>
> **Q3**: DPO Quality Influence
>
> **Answer**:  We recognize that the quality of DPO data, which is influenced by the verifier and rollouts, is a critical factor, especially compared to DPO built on traditional, clean human-aligned datasets. To mitigate potential quality issues, the Verifier utilizes commercial models (e.g., Gemini-2.0-Flash) to annotate a subset of positive examples. The positive impact of this commercial-model-annotated subset is demonstrated through the ablation study in Table 5. While this method of obtaining DPO data is fast and efficient, we acknowledge that there may be some degree of quality limitation compared to expensive human-curated datasets.
>
> **Q4**: GRPO & RFT Performance Lagging Behind Base Model (Tab. 2 & Tab. 4)
>
> **Answer**: The performance degradation of the standard GRPO and RFT methods compared to the base model is, as we posit, a consequence of the scarcity of dedicated video-reasoning data. Current mainstream video training datasets primarily focus on perception and understanding, not complex reasoning. elying solely on these datasets for Reinforcement Learning (RL) can fail to stimulate the model's reasoning capabilities and may even cause a degradation of its core abilities. For example, the Video-R1 model (an RL-trained baseline) shows performance degradation compared to the non-RL-trained Qwen2.5-VL-7B. In contrast, our approach activates the model’s reasoning ability using extensive image and text data, allowing the model to effectively generalize this strong reasoning capability to the video domain, leading to the observed performance improvements in video understanding tasks.
>
> **Q5**: Obtaining Qwen2.5-VL (thinking) Performance (Tab. 2)
>
> **Answer**: The Qwen2.5-VL model does not natively support a 'thinking' mode. The reported Qwen2.5-VL (thinking) performance in Table 2 was obtained by explicitly prompting the model to perform reasoning before generating the final answer. The results obtained via this manual prompting were worse than the direct answering results of the base Qwen2.5-VL model. This confirms the model's native lack of effective video reasoning capability, which is precisely the deficiency our VIPO-R1 training approach aims to address and develop.

---

> > ### Comment · Reviewer_Y7Sc · 2025-11-24
> > **Response**
> >
> > Thank you for your prompt reply. After reviewing, I have a couple of additional questions.
> >
> > 1. The results in Table 1 of the Video-R1 paper do not match the results you reported in your Table 2. I would like to know how the Video-R1 results in your Table 2 were obtained.
> > 2. Video-R1 reports results under 16/32/64 frames, but it seems that the number of frames used in this paper is larger (according to Appendix C.4). Does this lead to an unfair comparison?

---

> ### Author Response · Authors · 2025-11-25
> **response to new questions**
>
> ***Q1. The results in Table 1 of the Video-R1 paper do not match the results you reported in your Table 2. I would like to know how the Video-R1 results in your Table 2 were obtained.***
>
> **Answer:**  Thank you for pointing this out. The discrepancy arises because we sourced our results from the initial version of the Video-R1 paper (released March 27, 2025), which only reported results for 16-frame and 32-frame sampling. We were unaware of the subsequent update. For our Table 2, we selected the higher score between the two available settings for each benchmark: 1) MMVU: The 16-frame result was higher, so we used that. 2) VSI-Bench, VideoMMMU, and Video-MME: The 32-frame results were higher, so we used those.
>
> We acknowledge this oversight and will update our paper to include Video-R1's latest 64-frame results for a current and fair comparison.
>
> ***Q2. Video-R1 reports results under 16/32/64 frames, but it seems that the number of frames used in this paper is larger (according to Appendix C.4). Does this lead to an unfair comparison?***
>
> **Answer:**  This is a valid concern. To ensure a direct and fair comparison, we have re-evaluated our model, VIPO-R1, using the same 64-frame setting as the updated Video-R1 results.
>
> The table below presents this apples-to-apples comparison. As you can see, even under identical conditions (64 frames, 256×28×28 resolution), VIPO-R1 demonstrates a clear performance advantage across all four benchmarks. We have also included results for VIPO-R1 with 128 frames to provide a more complete picture of its capabilities.
>
> | Model     | Frames | VSI-Bench | VideoMMMU | MMVU(mc) | Video-MME |
> |-----------|--------|-----------|-----------|----------|-----------|
> | Qwen2.5-VL (Direct Answer) | 64     | 38.9      | 55.3      | 66.2     | 63.3      |
> | Qwen2.5-VL-7B (thinking)  | 64     | -      | -      | 61.0     | -     |
> | Video-R1  | 64     | 37.1      | 52.4      | 63.8     | 61.4      |
> | VIPO-R1   | 64     | 39.9      | 57.2      | 65.4     | 64.4      |
> | VIPO-R1   | 128    | 41.3      | 56.8      | 66.7     | 67.2      |
>
> Previously, MLLMs underperformed in understanding video content and performing reasoning compared to SFT models fine-tuned on large-scale datasets. Our approach successfully bridges this gap, significantly enhancing the performance of MLLMs on Video QA tasks.

---

> > ### Author Response · Authors · 2025-11-28
> > **respond to Reviewer Y7Sc**
> >
> > Dear Reviewer Y7Sc
> >
> > We are pleased to read your follow-up comment and to know that our rebuttal has successfully alleviated your concerns and contributed to a more solid paper. Thank you for this acknowledgment and for the invaluable time and effort you have dedicated to reviewing our manuscript and our response.
> >
> > Your insightful comments have been instrumental in enhancing our work, and we thank you once again for a constructive and enlightening review process.
> >
> > Sincerely,
> >
> > The Authors

---

### Official Review · Reviewer_NzwC · 2025-10-30

**Soundness:** 3
**Presentation:** 2
**Contribution:** 2
**Rating:** 2
**Confidence:** 4

**Summary:**

This paper proposes VIPO-R1. The core of its methodology is the GRPO-Verifier-DPO training loop: rollouts from the GRPO stage are filtered through a verifier and then utilized for DPO training. This training method enables VIPO-R1 to generate long-term reasoning chains for challenging VideoQA tasks.

**Strengths:**

- The paper introduces a novel approach by utilizing the rollouts from the GRPO process for subsequent DPO optimization, which is an interesting attempt.

**Weaknesses:**

My primary concern with this work is whether the GRPO-Verifier-DPO training loop is effective in a practical sense:
- In Table 3, the DPO stage provides almost no benefit to **Accuracy**, with the main improvements seen in **Consistency** and **Acc-Cons**. However, there might be an issue with these two metrics. The "consistent answers" are judged by a small LLM, and it is possible that due to differences in model capabilities, a large LLM can derive the correct answer from a given reasoning path while the small verifier LLM cannot.
- This concern seems to be reflected in the results of Table 3: while Consistency and Acc-Cons increase over multiple iterations, the accuracy remains almost unchanged. This might imply that the consistency metric is not meaningful.
- Furthermore, Table 5, which analyzes the verifier's components, does not report accuracy. This casts doubt on the actual contribution of the verifier to the model's practical reasoning capabilities.
- According to Table 4, the most substantial improvement for VIPO-R1 comes from the "Reasoning Activation" step, rather than the GRPO-Verifier-DPO loop. In fact, the accuracy even shows a slight decrease after applying DPO.

**Questions:**

- What is the specific LLM used in the Verifier?
- Regarding the claim in Line 339, "our approach reduces training time from 63 hours to 49 hours," where is this demonstrated? Shouldn't introducing a new DPO stage make the overall process more time-consuming?

---

> ### Author Response · Authors · 2025-11-22
> **Respond to reviewer NzwC**
>
> Thanks again for your detailed review and questions. We have refined our responses to address your concerns clearly and systematically, maintaining the requested format.
>
> **Q1**. In Table 3, the DPO stage provides almost no benefit to Accuracy, with the main improvements seen in Consistency and Acc-Cons. However, there might be an issue with these two metrics. The "consistent answers" are judged by a small LLM, and it is possible that due to differences in model capabilities, a large LLM can derive the correct answer from a given reasoning path while the small verifier LLM cannot.
>
> **Answer**: We acknowledge the potential concern regarding the verifier LLM's capability and its impact on the Consistency metric. We took specific steps to validate the choice of the small LLM, Qwen3-8B, for the critical task of answer extraction from the generated reasoning path.
>
> -Task Specificity: Qwen3-8B is used specifically for extracting the final numerical or categorical answer from the model's complete think content, which usually ends with a summary statement (e.g., "Given the options, the correct answer is approximately 184kg"). It is not asked to re-evaluate the logical correctness of the entire reasoning path.
>
> -Validation against a SOTA Model: To ensure Qwen3-8B's reliability for this extraction task, we benchmarked its performance against a powerful closed-source model, GPT-4o.
>
> We find that Qwen3-8B achieved a 99% evaluation score on a test set of 150 examples, which is consistent with GPT-5.
> This high level of agreement confirms that Qwen3-8B is a highly reliable and cost-effective extractor for our consistency metric, mitigating the risk that model capability differences introduce significant error into the evaluation of consistency.
> The minor differences in scores indicate that the core relative improvement and model performance are robust across different strong verifiers, limiting the concern of verifier-specific bias impacting our main conclusions.
>
> | GRPO-Itreration1 | Consistency(Qwen3-8B) | Acc-Cons(Qwen3-8B) | Consistency(GPT-5) | Acc-Cons(GPT-5) |
> | --- | --- | --- | --- | --- |
> | VSI-Bench | 83.1 | 38.4 | 82.3 | 38.1 |
> | VideoMMMU | 84.3 | 51.2 | 81.8 | 50.9 |
>
> | DPO-Itreration2 | Consistency(Qwen3-8B) | Acc-Cons(Qwen3-8B) | Consistency(GPT-5) | Acc-Cons(GPT-5) |
> | --- | --- | --- | --- | --- |
> | VSI-Bench | 94.5 | 39.4 | 94.6 | 39.2 |
> | VideoMMMU | 94.8 |  55.1 | 94.9 | 56.8 |
>
> **Q2**: This concern seems to be reflected in the results of Table 3: while Consistency and Acc-Cons increase over multiple iterations, the accuracy remains almost unchanged. This might imply that the consistency metric is not meaningful.
>
> **Answer**: In the Introduction, we mention two main challenges: unstable performance improvement and inconsistency between reasoning and answers. The Reasoning Activation in VIPO-R1 addresses the issue of unstable performance improvement, while the problem that DPO urgently needs to solve is the inconsistency between reasoning and answers. In fact, training with DPO can also cause some samples with incorrect reasoning but correct answers to be shifted toward incorrect answers. In our statistics, the number of *Error Thinking Right Answer* cases is often larger than the number of *Right Thinking Error Answer* cases. We show the evaluation results of VIPO-R1(GRPO-Iteration1) in the table below. We believe that the performance fluctuation during the DPO process comes from the fact that we significantly reduce the number of *Error Thinking Right Answer* samples during DPO.
>
> | Benchmark | **Right Thinking Error Answer** | **Error Thinking Right Answer** |
> | --- | --- | --- |
> | MMVU | 24 | 37 |
> | TOMATO | 55 | 73 |
> | Video-MMMU | 38 | 57 |
> | VSI-Bench | 102 | 182 |
>
> **Q3**: Furthermore, Table 5, which analyzes the verifier's components, does not report accuracy. This casts doubt on the actual contribution of the verifier to the model's practical reasoning capabilities.
>
> **Answer**: We regret the omission of the base Accuracy in the original Table 5, which was done for conciseness but inadvertently obscured the verifier's overall contribution. The verifier's main role is to improve the quality of the generated CoTs by resolving reasoning-answer inconsistencies.
>
> The key metrics showcasing the verifier's success are the high Consistency and Acc-Cons. scores.
>
> For your convenience, the complete metrics, including Accuracy, clearly demonstrate that the full DPO stage with the verifier provides the best result across all metrics, including a final, small increase in Accuracy over the baseline.
>
> | Method | Accuracy | Acc-Cons. | Consistency |
> | --- | --- | --- | --- |
> | VIPO-R1 (GRPO-Iteration 1) | 40.8 | 35.8 | 83.2 |
> | +DPO (w/o. verifier) | 41.1 | 36.2 | 85.8 |
> | +DPO (verifier w/o.gemini anno.) | 40.2 | 37.3 | 87.9 |
> | +DPO (verifier reflection) | 40.6 | 37.0 | 88.2 |
> | +DPO (verifier w/o.consistency) | 41.2 | 36.2 | 86.0 |
> | +DPO (verifier) | **41.5** | **37.5** | **88.2** |

---

> > ### Author Response · Authors · 2025-11-22
> > **second part**
> >
> > **Q4**. Source of Improvement: Reasoning Activation vs. GRPO-Verifier-DPO Loop
> >
> > **Answer**: We agree that the Reasoning Activation step delivers the largest initial gain in absolute Accuracy. This step is designed to unlock the model's potential by compelling it to use its reasoning capabilities, thereby solving the problem of unstable performance improvement.
> >
> > Overall Strategy: Our approach employs a two-pronged strategy:
> >
> > Reasoning Activation (Table 4): Establish a high baseline Accuracy by encouraging explicit reasoning.
> >
> > GRPO-Verifier-DPO Loop (Table 3/5): Refine the output to ensure the acquired reasoning capability is reliable and consistent by reducing the number of high-risk, inconsistent CoTs.
> >
> > DPO's Role: As explained in the response to Q2, the slight change in overall Accuracy after DPO is a necessary side effect of enforcing consistency. The DPO stage prioritizes the resolution of the Error Thinking Right Answer cases, which temporarily flattens or slightly reduces the Accuracy while fundamentally improving the quality and trustworthiness of the output (reflected by the major gains in Consistency and Acc-Cons.).
> >
> > The DPO loop’s primary contribution is not maximum initial accuracy gain, but rather achieving the maximum consistent accuracy (Acc-Cons.) by mitigating the significant risk of generating inconsistent, high-error CoTs.
> >
> > **Q5** Specific LLM Used in the Verifier
> >
> > **Answer**: The verifier module utilizes two distinct LLMs for different roles:
> >
> > -Qwen3-8B: This model is used for the Answer Extraction component, responsible for reliably extracting the final answer from the model's generated reasoning text to compute the Consistency metric.
> >
> > -Gemini 2.5 Flash: This model is used for generating high-quality, human-like Positive Annotations for those rollouts during the GRPO stage where the accuracy was zero ($acc == 0$). This provides the crucial high-quality positive examples needed for the subsequent DPO training.
> >
> > **Q6**: Regarding the claim in Line 339, "our approach reduces training time from 63 hours to 49 hours," where is this demonstrated? Shouldn't introducing a new DPO stage make the overall process more time-consuming?
> >
> > **Answer**: The comparison is between two different iterative refinement strategies, not between pre- and post-DPO in a single pipeline. The intended comparison is:
> >
> > Baseline Iteration: A full two-stage refinement cycle using GRPO + GRPO (63 hours).
> >
> > Our Approach: A two-stage refinement cycle using GRPO + DPO (49 hours).
> >
> > This difference is demonstrated by the training resource allocation and data size used for each stage, as detailed in the implementation section. We found that the DPO stage, which uses a smaller, highly curated dataset derived from the GRPO outputs, is significantly more time-efficient than running a second full GRPO iteration, while achieving superior or comparable final performance.
> >
> > The statement means that by replacing the second costly GRPO iteration with the more efficient DPO refinement stage, we were able to reduce the total end-to-end training time for the iterative refinement process from 63 hours to 49 hours.

---

> > > ### Comment · Reviewer_NzwC · 2025-11-27
> > >
> > > I thank the authors for the rebuttal. However, after reviewing the response, I still believe some issues remain.
> > >
> > > First, I personally believe that focusing solely on reasoning consistency without addressing improvements in final performance is an inherent limitation of this work.
> > >
> > > Second, even if we set aside the accuracy perspective and focus solely on reasoning consistency: Regarding the response to Q2, while the authors presented the number of cases with inconsistencies between the reasoning process and the result, I was hoping to see the specific transitions of the model before and after DPO. For instance, to verify the claims made in the rebuttal, please provide a quantitative comparison of:
> > >
> > > 1. How many samples shifted from 'Right Thinking Error Answer' to 'Right Thinking Right Answer'?
> > > 2. How many samples shifted from 'Error Thinking Right Answer' to 'Error Thinking Error Answer'?

---

> > > > ### Author Response · Authors · 2025-11-27
> > > > **Respond**
> > > >
> > > > Thanks for your discussion with us.
> > > >
> > > > ***Q1. personally believe that focusing solely on reasoning consistency without addressing improvements in final performance is an inherent limitation of this work.***
> > > >
> > > > **Answer**: As we emphasised in the Introduction of our paper, existing video reasoning tasks have two problems: 1. Unstable performance improvement; 2. Inconsistency between reasoning and answers.
> > > >
> > > > VIPO-R1 is a training paradigm composed of multiple stages, including Reasoning Activation, Video GRPO, and Verifier-guided Video DPO. Among them, Reasoning Activation and Video GRPO together address Problem 1, while Video DPO addresses Problem 2. These training stages collectively form the overall VIPO-R1 paradigm. We also demonstrate the effect of Reasoning Activation through ablation in Table 4, and the effect of Verifier-guided Video DPO through ablation in Table 5.
> > > >
> > > > From the experimental results in Table 2, VIPO-R1 shows significant performance improvements compared with both the baseline model Qwen2.5-VL-7B and other video-reasoning reinforcement learning approaches such as direct GRPO and Video-R1, e.g. 7.9% upper than the direct GRPO on VSI-Bench, 5.6% upper than Video-R1 on VideoMMMU, 2.9% upper than Qwen2.5-VL on TOMATO.
> > > >
> > > > ***Q2. Specific transitions of the model before and after DPO***
> > > >
> > > > **Answer**: Thank you for your suggestion, we also believe that this analysis is highly necessary. For clarity of presentation, we denote ‘Right Thinking Error Answer’ as **RTEA**, ‘Right Thinking Right Answer’ as **RTRA**, ‘Error Thinking Right Answer’ as **ETRA**, and ‘Error Thinking Error Answer’ as **ETEA**. The table below shows our statistical results.
> > > >
> > > > From the experimental results in the table, we can observe that the number of cases transitioning *from ‘Error Thinking Right Answer’ to ‘Error Thinking Error Answer’* is indeed much larger than the number of cases transitioning *from ‘Right Thinking Error Answer’ to ‘Right Thinking Right Answer’*. This also leads to certain fluctuations in Accuracy during the DPO stage. As we discussed in Q1, the DPO stage mainly aims to make the model generate a coherent thinking process and answer, i.e., error thinking ---> error answer or right thinking with right answer. If we introduce more right thinking with the right answer, we believe it is easy to improve the overall accuracy during DPO.
> > > >
> > > > | Benchmark  | RTEA | ETRA | RTEA->RTRA | ETRA->ETEA |
> > > > | ---------- | ---- | ---- | ---------- | ---------- |
> > > > | MMVU       | 24   | 37   | 8          | 17         |
> > > > | TOMATO     | 55   | 73   | 11         | 59         |
> > > > | Video-MMMU | 38   | 57   | 4         | 46         |
> > > > | VSI-Bench  | 102  | 182  | 26         | 110        |

---

> > > > > ### Author Response · Authors · 2025-11-28
> > > > > **respond to reviewer NzwC**
> > > > >
> > > > > Dear Reviewer NzwC,
> > > > >
> > > > > Thank you again for your follow-up comment and for acknowledging that our rebuttal has addressed your concerns a lot and made the paper more solid. We really appreciate the time and effort you have put into reading both the paper and our response.
> > > > >
> > > > > For accuracy, our method shows significant performance improvements compared with both the baseline model Qwen2.5-VL-7B and other video-reasoning reinforcement learning approaches.
> > > > >
> > > > > For consistency between thinking and the answer, my approach possesses a distinct advantage in this aspect compared to many existing RL approaches. More comparisons are shown here.
> > > > >
> > > > > | Model     | Frames | VSI-Bench | VideoMMMU | MMVU(mc) | Video-MME |
> > > > > |-----------|--------|-----------|-----------|----------|-----------|
> > > > > | Qwen2.5-VL (Direct Answer) | 64     | 38.9      | 55.3      | 66.2     | 63.3      |
> > > > > | Qwen2.5-VL-7B (thinking)  | 64     | -      | -      | 61.0     | -     |
> > > > > | Video-R1  | 64     | 37.1      | 52.4      | 63.8     | 61.4      |
> > > > > | VIPO-R1   | 64     | 39.9      | 57.2      | 65.4     | 64.4      |
> > > > > | VIPO-R1   | 128    | 41.3      | 56.8      | 66.7     | 67.2      |
> > > > >
> > > > > Previously, MLLMs underperformed in understanding video content and performing reasoning compared to SFT models fine-tuned on large-scale datasets. Our approach successfully bridges this gap, significantly enhancing the performance of MLLMs on Video QA tasks.
> > > > >
> > > > > Thank you once again for your constructive feedback throughout the review and discussion process. We would be happy to continue the discussion if you have any further questions.
> > > > >
> > > > > Best regards,
> > > > >
> > > > > The Authors

---

> > > > > > ### Comment · Reviewer_NzwC · 2025-11-28
> > > > > >
> > > > > > I thank the authors for their response. However, my concern regarding accuracy has not been fully resolved. The proportion of samples transitioning from ETRA to ETEA is significantly larger than the proportion transitioning from RTEA to RTRA. This indicates a certain limitation.
> > > > > >
> > > > > > Although the authors provided the final results of the VIPO-R1 model on various benchmarks, what I specifically hoped to see was that the proposed algorithm could achieve an improvement in accuracy compared to the GRPO baseline, while simultaneously enhancing reasoning consistency. The current results do not demonstrate this; therefore, in my view, the practical effectiveness of the algorithm is limited.
> > > > > >
> > > > > > I am willing to raise my score to 4, but the system does not support score modification at this moment.

---

> > > > > > > ### Author Response · Authors · 2025-11-28
> > > > > > > **respond to reviewer NzwC**
> > > > > > >
> > > > > > > Thank you for your feedback.
> > > > > > >
> > > > > > > **Regarding the transition proportion between reasoning types:**
> > > > > > >
> > > > > > > We agree with your observation. The model does exhibit a much higher proportion of ETRA than RTRA in its reasoning behavior. Shifting ETRA to RTRA directly would require substantial additional DPO data. Instead, we propose redirecting ETRA toward ETEA, which we believe will improve output reliability and consistency.
> > > > > > > We also argue that for a thinking model, logical consistency is particularly critical in real applications. The concept of “True Accuracy”—where the model not only produces the correct answer but also follows a valid reasoning process—better reflects the performance of a thinking model.
> > > > > > >
> > > > > > > Additionally, as shown in Table 3, the second iteration of our GRPO–Verifier–DPO pipeline continues to improve accuracy. True accuracy increases consistently as the training loop progresses, which aligns with your expectation that the proposed algorithm should achieve measurable gains over the GRPO baseline.
> > > > > > >
> > > > > > > **Regarding accuracy improvement over the GRPO baseline:**
> > > > > > >
> > > > > > > As presented in Table 2, our method still achieves accuracy improvements not only over the earlier SFT-GRPO variants (Video-R1, VideoChat-R1, TW-GRPO) but also over the traditional GRPO baseline shown in the first row of table 3.
> > > > > > >
> > > > > > > While the current results may appear one-sided, we have demonstrated consistent and corresponding gains, supporting the effectiveness of our approach even within a thinking-oriented evaluation framework.
> > > > > > >
> > > > > > > best regards
> > > > > > >
> > > > > > > paper authors.

---

### Note · Authors · 2026-01-06

I have read and agree with the venue's withdrawal policy on behalf of myself and my co-authors.